# Weakly Supervised Discovery of Semantic Attributes

**Ameen Ali**                                                    AMEENALI@MAIL.TAU.AC.IL
*School of Computer Science*
*Tel Aviv University*
*Tel Aviv, Israel*

**Tomer Galanti**                                                  GALANTI@MIT.EDU
*Center for Brains, Minds and Machines (CBMM)*
*Massachusetts Institute of Technology*
*Cambridge, MA, USA*

**Evgenii Zheltonozhskii**                          EVGENIIZH@CAMPUS.TECHNION.AC.IL
*Department of Computer Science*
*Technion – Israel Institute of Technology*
*Haifa, Israel*

**Chaim Baskin**                                 CHAIMBASKIN@CS.TECHNION.AC.IL
*Department of Computer Science*
*Technion – Israel Institute of Technology*
*Haifa, Israel*

**Lior Wolf**                                                    WOLF@CS.TAU.AC.IL
*School of Computer Science*
*Tel Aviv University*
*Tel Aviv, Israel*

**Editors:** Bernhard Schölkopf, Caroline Uhler and Kun Zhang

## Abstract

We consider the problem of extracting semantic attributes, using only classification labels for supervision. For example, when learning to classify images of birds into species, we would like to observe the emergence of features used by zoologists to classify birds. To tackle this problem, we propose training a neural network with discrete features in the last layer, followed by two heads: a multi-layered perceptron (MLP) and a decision tree. The decision tree utilizes simple binary decision stumps, thus encouraging features to have semantic meaning. We present theoretical analysis, as well as a practical method for learning in the intersection of two hypothesis classes. Compared with various benchmarks, our results show an improved ability to extract a set of features highly correlated with a ground truth set of unseen attributes.

**Keywords:** Feature discovery, quantization, explainability

## 1. Introduction

The discovery of meaningful intermediate features in classification problems is at the heart of many scientific domains. For example, in botany, species are often identified based on dichotomous keys related to the shape of the leaf, the texture of the fruit, etc. (Project, 2021). In paleography, manuscripts are dated or attributed to a specific scribe based, among other attributes, on specific characteristics of the morphology of the letters (Stokes, 2009).

Such intermediate features allow experts to discuss a certain classification outcome, for example, in case of an ambiguity or disagreement. Moreover, the attributes discovered help understand the underlying structure of the problem. In our paleography example, certain morphological attributes can be traced back to a specific school of scribes, and certain forms evolve over time in specific geographic locations, influenced by nearby writing cultures (James, 2020).

In this work, we define a weakly supervised attribute discovery problem, motivated by the nature of such intermediate level features. The problem is addressed within a novel theoretical framework, which gives rise to an effective method. Our focus is on binary attributes that are both evidence-based and distinctive. By evidence-based we mean that there is a mapping $f$ such that every input $x$ is mapped to a vector $f(x)$ of binary values, indicating the presence of each attribute. Distinctiveness means that there should be simple rules that determine the class label $y(x)$ based on the obtained attributes $f(x)$.

Currently, deep neural networks (DNNs) represent the most successful methodology for obtaining attributes from images (Zhang et al., 2014; Xu et al., 2020) and for classification. However, interpreting DNNs is very challenging (Zhang et al., 2021). In contrast, decision trees with binary decision stumps provide simple and interpretable decision rules, which are based on attributes (Guidotti et al., 2018). We propose merging the two approaches by training a DNN to produce quantized representations suitable for classification both by an MLP and by a decision tree.

To perform this hybrid learning task, we provide a theoretical analysis of the problem of learning at the intersection of two hypothesis classes. We study the optimization dynamics of training two hypotheses from two different classes of functions. The first one is trained to minimize its distance from the second one, and the second one is trained to minimize its distance from the first one and the target function. We call this process *Intersection Regularization*, since it regularizes the second hypothesis to be close to the first hypothesis class. We discuss the conditions on the loss surface in terms of theory, for which this process converges to an equilibrium point or a local minimum.

We present a method for concurrently training a network and a tree, based on the proposed analysis. The algorithm learns a quantized representation of the data and two classifiers on top of this representation: a decision tree and an MLP. The two classifiers are trained for a supervised multiclass classification task using intersection regularization. Our goal is to recover attributes that match those that humans would provide, without having access to such attributes during training. Since the human-defined attributes are typically sparse, we also apply $L_1$ regularization on the quantized vector of activations.

In an extensive set of experiments, we demonstrate that discrete representations, along with decision stumps learned using our algorithm, are highly correlated with a set of unseen human-defined attributes. At the same time, the overall classification accuracy is only slightly reduced compared to standard cross-entropy training.

The key contributions of this paper are as follows: (i) We identify a novel learning problem, which we call *Weakly Unsupervised Discovery of Semantic Attributes*. In this setting, samples are associated with abstract binary attributes and are labeled by class membership. The algorithm is provided with class-labeled samples and is assessed based on its ability to recover the binary attributes without any access to them. We provide concrete measures for testing the success of a given method for this task. (ii) We introduce a method for recovering the semantic attributes. This method is based on a novel regularization process for regularizing a neural network using a decision tree. We call this regularization method, *'Intersection Regularization'*, and (iii) We study the theory of convergence guarantees of the new regularization method to equilibrium points and local minima.

## 2. Related Work

**Interpretability** Developing tools and techniques to interpret existing deep learning based approaches and to build explainable machine learning algorithms is a fast-growing field of research. In computer vision, most contributions are concerned with providing an output relevance map. These methods include saliency-based methods (Simonyan et al., 2013; Zeiler and Fergus, 2014; Mahendran and Vedaldi, 2016; Zhou et al., 2016a; Dabkowski and Gal, 2017; Zhou et al., 2018; Gur et al., 2020), Activation Maximization (Erhan et al., 2009) and Excitation Backprop (Zhang et al., 2018), perturbation-based methods Fong and Vedaldi (2017); Fong et al. (2019). Shapley-value-based methods (Lundberg and Lee, 2017) enjoy theoretical justification and the Deep Taylor Decomposition (Montavon et al., 2017) provides a derivation that is also applicable to Layer-wise Relevance Propagation (LRP) (Bach et al., 2015) and its variants (Gu et al., 2018; Iwana et al., 2019; Nam et al., 2019) presented RAP. Gradient-based methods are based on the gradient with respect to the layer's input feature map and include Gradient*Input (Shrikumar et al., 2016), Integrated Gradients (Sundararajan et al., 2017), SmoothGrad (Smilkov et al., 2017), FullGrad (Srinivas and Fleuret, 2019), and Grad-CAM (Selvaraju et al., 2017).

Methods that provide an output relevance map suffer from several notable disadvantages. First, many of these methods were shown to suffer from a bias toward image edges and fail sanity checks that link their outcome to the classifiers, as shown by Asano et al. (2019). Furthermore, even though these methods are useful for visualization and downstream tasks, such as weakly supervised segmentation (Li et al., 2018), it is not obvious how to translate the image maps produced by these methods into semantic attributes, i.e., extract meaning from a visual depiction. An additional disadvantage of this approach is that it does not provide a direct method for evaluating whether a given neural network is interpretable or not, and the evaluation is often done with related tasks, such as segmentation or measuring the classifier's sensitivity to regions or pixels deemed important to the classification outcome or the absence of this. In our framework, we suggest objective measures in which one can quantify the degree of interpretability of a given model.

Since linear models are intuitively considered interpretable, Alvarez-Melis and Jaakkola (2018a) developed a framework in which the learned features are monotonic and additive. The explanation takes the form of presenting each attribute's contribution, explaining the attributes using prototype samples. Unlike our framework, no attempt was made to validate that the obtained attributes correspond to a predefined list of semantic attributes.

Local interpretability models, such as LIME (Ribeiro et al., 2016), approximate the decision surface for each specific decision by a linear model and, unlike our method, are not aimed at extracting meaningful attributes. Besides, such models are known to be sensitive to small perturbations of the input (Alvarez-Melis and Jaakkola, 2018b; Yeh et al., 2019).

**Fine-Grained Classification** Fine-grained classification aims at differentiating subordinate classes of a common superior class. Those subordinate classes are usually defined by domain experts, based on complicated rules, which typically focus on subtle differences in particular regions. Because of the inherent difficulty of classifying slightly different classes, many contributions in this area often aim at detecting informative regions in the input images that aid their classification (Duan et al., 2012; Yang et al., 2018; Chen et al., 2019; Hu et al., 2019; Huang and Li, 2020; Zhuang et al., 2020). In our work, we would like to recover semantic attributes that are not necessarily associated with specific image regions, e.g., mammal vs. reptile, or omnivore vs. carnivore.

**Disentanglement** A disentangled representation (Schmidhuber, 1992; Bengio et al., 2013a; Peters et al., 2017; Lake et al., 2017; Tschannen et al., 2018) is a representation that contains multiple independent parts. Various methods, such as (Brakel and Bengio, 2017; Feng et al., 2018; Marx et al., 2019; Tsai et al., 2020) have been able to effectively learn disentangled representations from data. This setting is different from ours in two ways. First, the attributes we would like to recover are not necessarily independent. Second, in disentanglement, we are typically interested in recovering any set of features that represent the data in a disentangled manner. In our case, success is measured subject to a specific set of attributes.

**Attributes Discovery** Attributes discovery has been an active research direction in computer vision. Attributes are typically referred as human-interpretable features that can describe the input image or the class it belongs to. Gutierrez et al. (2019) cast the problem of attributes discovery as a supervised learning task. Lampert et al. (2009); Farhadi et al. (2009a); Yu et al. (2013); Sattar et al. (2017) focus on category-level attributes, in which one would like to learn a set of attributes that describe the classes of the different images in the dataset. In several cases, and especially in fine-grained classification, predicting the class of an image is a more nuanced process, where the values of attributes may vary between images in the same category. In our paper, we focus on recovering instance-level attributes.

**Explainability of Decision Trees** Decision trees are known to be naturally explainable models, as long as the decision rules are easy to interpret. However, as far as we know, it has not been shown before that learning trees, while optimizing the features, lead to the emergence of attributes that are similar to those that appear in large set of (ground truth) relevant attributes.

In the context of the interpretability of recommendation systems, meta-trees were used for providing per-user decision rules (Shulman and Wolf, 2020). This method relies on a pre-existing set of features rather than extracting them from the input. In another line of work, Frosst and Hinton (2017) also employs preexisting features and distills the information within a deep network into a soft decision tree (Irsoy et al., 2012). Unlike our work, the neural network is not optimized to produce suitable features.

More relevant to our work is the Adaptive Neural Trees method of Tanno et al. (2019), which also combines decision trees and neural networks. In this method, trees of dynamic architectures that include network layers are grown so that the underlying network features gradually evolve. It is shown that the learned tree divides the classes along meaningful axes; however, the emergence of semantic attributes was not shown.

**Quantization** Quantization is the conversion of floating-point data to a low-bit discrete representation. The most common approaches for training a quantized neural network employ two sets of weights (Hubara et al., 2018; Zhou et al., 2016b). The forward pass is performed with quantized weights, and updates are performed on full precision ones, i.e., approximating gradients with the straight-through estimator (STE, Bengio et al., 2013b). In this work, we make use of quantizations in order to learn discrete, interpretable representations of the data within a neural network.

## 3. Problem Setup

In this section, we formulate a new learning setting, which we call *Weakly Unsupervised Discovery of Semantic Attributes*. In this setting, there is an unknown target function $y : \mathcal{X} \rightarrow \mathcal{Y}$, along with an attributes function $f : \mathcal{X} \rightarrow \mathcal{U}^d$ we would like to learn. Here, $\mathcal{X} \subset \mathbb{R}^n$ is a set of instances and $\mathcal{Y}$ is

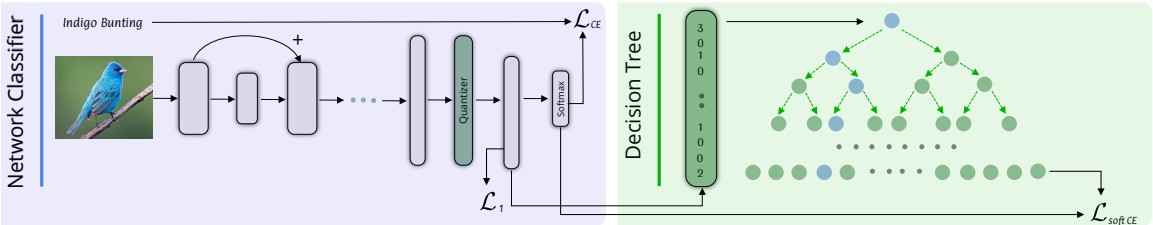

Figure 1: **An illustration of our method.** The model has three main components: **(a)** a quantized representation function $F_q$, **(b)** a classifier neural network $G$ and **(c)** a multivariate decision tree $T$. We have three losses: a $L_1$ regularization loss over the masked quantized features, a cross-entropy loss between the network and the ground-truth labels, and a soft cross-entropy loss between the outputs of the tree and the network.

a set of labels. The function $f$ is assumed to be $d_0$-sparse, i.e., $\forall x \in \mathcal{X} : \|f(x)\|_0 \leq d_0$. The values of $f$ are taken from a latent space $\mathcal{U}^d$, where $\mathcal{U} = \mathbb{R}$, $[-1, 1]$ or $\{\pm 1\}$, depending on the task.

For example, $\mathcal{X}$ can be a set of images of $k$ animal species and $\mathcal{Y} = \{e_i\}_{i=1}^k$ the set of labels, where $e_i \in \mathbb{R}^k$ is the $i$'th elementary vector. In this case, $y$ is a classifier that takes an animal image $x$ and returns the species of the animal. The function $f$ corresponds to a set of $d$ binary attributes $\{f_i\}_{i=1}^d$ (e.g., carnivore/herbivore). Each attribute $f_i$ takes an image $x$ and returns whether the animal illustrated in $x$ satisfies the $i^{\text{th}}$ attribute.

Similarly to the standard learning setting, the set $\mathcal{X}$ is endowed with a distribution $D$, from which samples $x$ are taken. The learning algorithm is provided with a set $\mathcal{S} = \{(x_i, y(x_i))\}_{i=1}^m$ of $m$ i.i.d. samples. Typically, the learning algorithm is supposed to fit a hypothesis $H : \mathcal{X} \to \mathcal{Y}$ from a given *hypothesis class* $\mathcal{H}$ that would minimize the *expected risk*:

$$\mathcal{L}_D[H, y] := \mathbb{E}_{x \sim D}[\ell(H(x), y(x))], \tag{1}$$

where $\ell : \mathcal{Y}^2 \to [0, \infty)$ is a loss function, for example, the $\ell_2$ loss function $\ell_2(a, b) = \|a - b\|_2^2$ for regression or the cross-entropy loss for classification. Since the algorithm is not provided with full access to $D$, it minimizes the empirical version of it

$$\mathcal{L}_\mathcal{S}[H, y] := \frac{1}{m} \sum_{(x, y(x)) \in \mathcal{S}} \ell(H(x), y(x)), \tag{2}$$

or minimizes $\mathcal{L}_\mathcal{S}[H, y]$ along with additional regularization terms.

In contrast to the standard learning setting, in our framework we would like to learn a model $H = G \circ q \circ F$ that minimizes the expected risk, but also recovers the semantic attributes $f(x)$ (without any access to these attributes). Here, $F$ and $G$ are a trainable representation function and a classifier, and $q$ is a pre-defined discretization operator.

Put differently, the labels are given as proxy labels, while the actual goal is to learn a discrete-valued representation $q(F(x))$ of the data, which maximizes some measure of feature fidelity $d_D(F) = d_D(f; b \circ q \circ F)$ depending on the distribution $D$ and a binarization function $b$ that translates a vector into a binary vector (see Sec. 5 for details).

### 3.1. Measures of Feature Fidelity

The goal of training in the proposed setting is to learn a representation of the data $q(F(x))$ that maximizes the fidelity of the extracted features with respect to an unseen set of ground-truth binary attributes $f(x)$.

In this section, we define a generic family of functions $d_D(f; g)$ for measuring the fidelity of a multi-variate function $g : \mathcal{X} \to \mathcal{U}^d$ with respect to a multi-variate function $f : \mathcal{X} \to \mathcal{U}^n$ over a distribution $D$ of samples.

Let $r(q_1, q_2; D)$ be a measure of accuracy between two univariate functions $q_1, q_2 : \mathcal{X} \to \mathcal{U}$ over a distribution $D$. In this paper, since we consider imbalanced attributes, $r$ is the F1 score. Finally, we extend $r$ to be annotation-invariant by using $\hat{r}(q_1, q_2; D) := \max\{r(q_1, q_2; D), r(q_1, 1 - q_2; D)\}$ as a measure of accuracy that is invariant to whether positive samples are denoted by 1 or 0.

Let $g : \mathcal{X} \to \{0, 1\}^d$ and $f : \mathcal{X} \to \{0, 1\}^n$ be two multivariate binary functions. We denote

$$d_D(f\|g) := \max_{\pi:[n]\to[d]} \frac{1}{n}\sum_{i=1}^{n} \hat{r}(f_i, g_{\pi(i)}; D) = \frac{1}{n}\sum_{i=1}^{n} \max_{j\in[d]} \hat{r}(f_i, g_j; D). \tag{3}$$

This quantity measures the average similarity between each feature $f_i$ in $f$ to some feature $g_j$ in $g$.

The fidelity of $g$ with respect to $f$ is the harmonic mean $d_D(f; g) := 2\frac{d_D(f\|g)\cdot d_D(g\|f)}{d_D(f\|g)+d_D(g\|f)}$, which is a symmetric measure of similarity. Informally, $d_D(f\|g)$ measures the extent at which the set of attributes in $f$ can be treated as a subset of the set of attributes in $g$ and $d_D(f; g)$ as the extent at which the sets of attributes in $f$ and $g$ are equivalent. We use the harmonic mean as it penalizes low values, in contrast to the arithmetic mean.

When measuring the fidelity over a finite set of test samples $\mathcal{S}_T$, the proposed $d_D(f; g)$ considers the discrete uniform distribution $D = U[\mathcal{S}_T]$.

Finally, in some cases, it is more relevant to measure the distance between $f$ and $g$ instead of their similarity. We note that our method can be readily extended to this case by taking $r$ to be a distance function and replacing the maximization in Eq. (3) and the definition of $\hat{r}$ by minimization. For example, for real-valued functions (i.e., $\mathcal{U} \subset \mathbb{R}$), it is reasonable to use the distance between the two functions, $r(q_1, q_2; D) = \mathbb{E}_{x\sim D}[|q_1(x) - q_2(x)|]$.

## 4. Intersection Regularization

A key component of our method in Sec. 5 is the proposed notion of *Intersection Regularization*. In this section, we introduce and study the optimization dynamics of the intersection regularization theoretically.

Suppose that we have two hypothesis classes $\mathcal{G} = \{G_\theta \mid \theta \in \Theta\}$ and $\mathcal{T} = \{T_\omega \mid \omega \in \Omega\}$, where $\Theta$ and $\Omega$ are two sets of parameters. Intersection regularization involves solving the following problem

$$\min_{\theta\in\Theta} \min_{\omega\in\Omega} \mathcal{Q}(\theta, \omega)$$
$$\text{where } \mathcal{Q}(\theta, \omega) := \mathcal{L}_{\mathcal{S}}[G_\theta, y] + \mathcal{L}_{\mathcal{S}}[G_\theta, T_\omega]. \tag{4}$$

In this problem, we are interested in learning a hypothesis $G_\theta \in \mathcal{G}$ that is closest to the target function $y$ among all members of $\mathcal{G}$ that can be approximated by a hypothesis $T_\omega \in \mathcal{T}$. We can think of $\mathcal{T}$ as prior knowledge we have on the target function $y$. Therefore, in some sense, the term $\mathcal{L}_{\mathcal{S}}[G_\theta, T_\omega]$ acts as a regularization term that restricts $G_\theta$ to be close to the class $\mathcal{T}$.

In Sec. 5, we use intersection regularization to train a neural network $G$ over a quantized representation that mimics a decision tree $T$ and minimizes the classification error. In this case, the class $\mathcal{T}$ consists of decision trees of a limited depth ($\omega$ is a vector that allocates an encoding of the tree, including its structure and decision rules) and $\mathcal{G}$ is a class of neural networks of a fixed architecture ($\theta$ is a vector of the weights and biases of a given network). The underlying quantized representation thus obtained is suitable for classification by both a decision tree and a neural network.

The following analysis focuses on two main properties: finding a local minima of $\mathcal{Q}$ and arriving at an equilibrium point of $\mathcal{Q}$. An equilibrium point of $\mathcal{Q}$ is a pair $(\theta, \omega)$, such that, $\mathcal{Q}(\hat{\theta}, \hat{\omega}) = \min_\theta \mathcal{Q}(\theta, \hat{\omega}) = \min_\omega \mathcal{Q}(\hat{\theta}, \omega)$. The proofs are given in the appendix and are based on Thm. 2.1.14 by Nesterov (2014) and the analysis by Song et al. (2017).

The following proposition shows that under certain conditions it is possible to converge to an equilibrium of $\mathcal{Q}(\theta, \omega)$ when iteratively optimizing $\theta$ and $\omega$. To show this, we assume that $\mathcal{Q}$ is a convex function with respect to $\theta$ for any fixed value of $\omega$. This is true, for example, when $G_\theta$ is the linearization of a wide neural network (Lee et al., 2019), which also serve as universal approximators (Ji et al., 2020). It has also been proven that the optimization dynamics of wide neural networks match the dynamics of their linearized version (Lee et al., 2019). In addition, we assume that one is able to compute a global minimizer $\omega$ of $\mathcal{L}_\mathcal{S}[G_\theta, T_\omega]$ for any $\theta$. This is typically impossible, however, it is reasonable to assume that one is able to approximately minimize $\mathcal{L}_\mathcal{S}[G_\theta, T_\omega]$ with respect to $\omega$ if it is being optimized by a descent optimizer. Throughout the analysis, we assume that $\cup_{\omega \in \Omega} \arg\min_\theta \mathcal{Q}(\theta, \omega)$ is well-defined and bounded and that $\lim_{\theta: \|\theta\| \to \infty} \mathcal{L}_\mathcal{S}[G_\theta, y] = \infty$.

**Proposition 1** *Assume that $\mathcal{Q}(\theta, \omega)$ is convex and $\beta$-smooth w.r.t $\theta$ for any fixed value of $\omega$. Let $\theta_1$ be an initialization and $\omega_1 \in \arg\min_\omega \mathcal{Q}(\theta_1, \omega)$. We define $\theta_t$ to be the weights produced after $t$ iterations of applying Gradient Descent on $\mathcal{Q}(\theta, \omega_{t-1})$ over $\theta$ with learning rate $\mu < \beta^{-1}$ and $\omega_t = \arg\min_\omega \mathcal{Q}(\theta_{t-1}, \omega)$. Then, we have*

$$\lim_{t \to \infty} \mathcal{Q}(\theta_t, \omega_t) = \lim_{t \to \infty} \min_\theta \mathcal{Q}(\theta, \omega_t) = \lim_{t \to \infty} \min_\omega \mathcal{Q}(\theta_t, \omega).$$

The following proposition shows that if we apply Block Coordinate Gradient Descent (BCGD) in order to optimize $\theta$ and $\omega$ minimize $\mathcal{Q}(\theta, \omega)$ (starting at $(\theta_1, \omega_1)$), then they converge to a local minimum that is also an equilibrium point. The BCGD iteratively updates: $\theta_{t+1} = \theta_t - \mu \nabla \mathcal{Q}(\theta_t, \omega_t)$ and $\omega_{t+1} = \omega_t - \mu \nabla \mathcal{Q}(\theta_{t+1}, \omega_t)$. Throughout the analysis we assume that the sets $\cup_{\omega \in \Omega} \arg\min_\theta \mathcal{Q}(\theta, \omega)$ and $\cup_{\theta \in \Theta} \arg\min_\omega \mathcal{Q}(\theta, \omega)$ are well-defined and bounded.

**Proposition 2** *Assume that $\mathcal{Q}(\theta, \omega)$ is a twice continuously differentiable, element-wise convex (i.e., convex w.r.t $\theta$ for any fixed value of $\omega$ and vice versa), Lipschitz continuous and $\beta$-smooth function, whose saddle points are strict. Let $\theta_t, \omega_t$ be the weights produced after $t$ iterations of applying BCGD on $\mathcal{Q}(\theta, \omega)$ with learning rate $\mu < \beta^{-1}$. $(\theta_t, \omega_t)$ then converges to a local minimum $(\hat{\theta}, \hat{\omega})$ of $\mathcal{Q}$ that is also an equilibrium point.*

## 5. Method

In this section, we propose our method for dealing with the problem introduced in Sec. 3.

---

**Algorithm 1** Intersection Regularization-based Sparse Attributes Discovery

---

**Require:** $\mathcal{S} = \{(x_i, y(x_i))\}_{i=1}^m$ - dataset; $\lambda_1, \lambda_2, \lambda_3$ - non-negative coefficients; $I$ - number of epochs; $s$ - batch size; $A$ - Tree training algorithm (and splitting criteria);

1: Initialize $F, G$ and $T = $ None;
2: Partition $\mathcal{S}$ into batches $Batches(\mathcal{S})$ of size $s$;
3: **for** $i = 1, \ldots, I$ **do**
4:    $\bar{\lambda}_2 = \mathbb{1}[i > 1] \cdot \lambda_2$;
5:    $\mathcal{S}' = \emptyset$;
6:    **for** $B \in Batches(\mathcal{S})$ **do**
7:       Update $G$ using GD to minimize $\lambda_1 \mathcal{L}_B[G \circ F_q, y] + \bar{\lambda}_2 \mathcal{L}_B[G \circ F_q, T \circ F_q]$;
8:       Update $F$ using GD to minimize $\lambda_1 \mathcal{L}_B[G \circ F_q, y] + \bar{\lambda}_2 \mathcal{L}_B[G \circ F_q, T \circ F_q] + \lambda_3 \mathcal{R}_B[F]$;
9:       Extend $\mathcal{S}' = \mathcal{S}' \cup \{(F_q(x), G(F_q(x))) \mid x \in B\}$;
10:   **end for**
11:   Initialize decision tree $T$;
12:   Train $T$ over $\mathcal{S}'$ using $A$;
13: **end for**
14: **return** $F, G, T$;

---

**Model**  Informally, our algorithm aims to learn a sparse discrete representation $F_q(x) := q(F(x))$ of the data, that is suitable for classification by a decision tree $T$ of small depth. Intuitively, this leads to a representation that supports classification by relatively simple decision rules. For this purpose, we consider a model of the following form: $H_{tree} = T \circ F_q$, where $F : \mathcal{X} \to \mathbb{R}^d$ is a trainable neural network and $T : \mathbb{R}^d \to \mathbb{R}^k$ is a decision tree from a class $\mathcal{T}$ of multivariate regression decision trees of maximal depth $d_{\max}$. The function $G$ (and $T$) is translated into a classifier by taking $\arg\max_{i \in [k]} G(F_q(x))$ (and $\arg\max_{i \in [k]} T(F_q(x))$).

In order to learn a neural network $G$ that minimizes the classification error and also approximates a decision tree, we apply intersection regularization between a class of neural networks $\mathcal{G}$ and the class of decision trees $\mathcal{T}$. Our objective function is decomposed into several loss functions. For each loss, we specify in brackets "$[ \cdot ]''$ the components responsible for minimizing the specified loss functions. For a full description of our method, see Alg. 1.

For each epoch (line 3), we iteratively update the network to minimize its objective function using GD and train the decision tree from scratch. To optimize the neural network, we use two loss functions (lines 7-8). The first one is the cross-entropy loss of $H_{net}$ with respect to the ground-truth labels, and the second one is the soft cross-entropy loss of $H_{net}$ with respect to the probabilities of $H_{tree}$,

$$\mathcal{L}_B[G \circ F_q, y] \text{ and } \mathcal{L}_B[G \circ F_q, T \circ F_q], \tag{5}$$

where the loss function $\ell : \Delta_k \times \Delta_k \to [0, \infty)$ is the cross-entropy loss defined as follows: $\ell(u, v) := -\sum_{i=1}^k v_i \cdot \log(u_i)$, $k$ is the number of classes and $\Delta_k$ is the standard simplex. The second loss is applied only from the second epoch onwards (line 4).

To learn sparse representations (Tibshirani, 1996; Koh et al., 2007), we also apply $L_1$ regularization over the quantized representation of the data

$$\mathcal{R}_B[F] := \frac{1}{m} \sum_{i=1}^s \|F_q(x_i)\|_1. \tag{6}$$

**Tree Optimization**    During each epoch, we accumulate a dataset $\mathcal{S}'$ of pairs $(F_q(x), G(F_q(x)))$ for all of the samples $x$ incurred until the current iteration (line 9 in Alg. 1). By the end of the epoch, we train a multivariate regression decision tree $T$ from scratch over the dataset $\mathcal{S}'$ (see line 12 in Alg. 1). To train the decision tree, we used the CART algorithm of Loh (2011) with the information gain splitting criteria.

**Quantization and Binarization**    The function $F : \mathcal{X} \to \mathbb{R}^n$ is a real-valued multivariate function. To obtain a discrete representation of the data, we discretize the outputs of $F$ using the uniform quantizer (Sheppard, 1897; Vanhoucke et al., 2011) $q$, see appendix B for a step-by-step listing of the algorithm. This quantization employs a finite number of equally sized bins. Their size is calculated by dividing the input range into $2^r$ bins, where $r$ specifies the number of bits for encoding each bin.

Since the round function is non-differentiable, the gradients of the uniform quantization are usually approximated (Bengio et al., 2013b; Ramapuram and Webb, 2019; Yang et al., 2019). In this paper, we utilize the straight-through estimator (Bengio et al., 2013b), which assumes that the Jacobian of rounding is simply identity, to estimate gradients of the discretization.

For applying the feature fidelity measure proposed in Sec. 3.1, we first cast the discrete vector $q(F(x))$ into a vector of binary features $b(q(F(x)))$ and compute $d_D(F) := d_D(f; b \circ q \circ F)$, where $b$ is a binarization function. In this paper, we use the following binarization scheme. For a given vector $v = q(F(x))$ we compute $u = b(v) := (u^1 \| u^2)$ as follows: for all $i \in [n]$ and $j \in [2^r]$, we have: $u^1_{i,j} = \mathbb{1}[v_i = j]$ and $u^2_{i,j} = \mathbb{1}[v_i \neq j]$, where $\mathbb{1}$ is the indicator function. The dimension of $u$ is $2^{r+1} \cdot n$.

## 6. Experiments

In this section, we evaluate our method on several datasets with attributes in comparison to various baselines.

**Implementation Details**    The architecture of the feature extractor $F$ is taken from (Du et al., 2020) with the published hyperparameters and is based on ResNet-50 . The classifier $G$ is a two-layered, fully connected neural network. We initialize the ResNet with pre-trained weights trained on ILSVRC2012 (Russakovsky et al., 2015). Finetuning a pre-trained model, instead of training the model from scratch is the common practice in fine-grained classification, for example Du et al. (2020); Zhuang et al. (2020) employ an imagenet trained backbone model and finetune it over the fine-grained classification task. If one trains any of these models (including ours) on fine-grained classification tasks without pre-training the model, the results for the classification accuracy will be lower, and we can expect the fidelity score to be lower, as well. When training our method and the baselines we employed the following early stopping criterion: we train the model until the second epoch for which the accuracy rate drops over a validation set, and report the results for the last epoch before the drop. In order to conduct a fair comparison we matched the original scheduler of Li et al. (2018)

Throughout the experiments, we used the following default hyperparameters, except in our ablation studies, where we varied the hyperparameters in order to evaluate their effect. The coefficients for the loss functions are $\lambda_1 = 2$, $\lambda_2 = 1$, $\lambda_3 = 0.001$. Optimization was carried out using SGD. We used batch size of 64 across all of the experiments, and $0.0001$ as the learning rate . For the data augmentations, RandomHorizontalFlip,RandomCrop are used with flipping probability $p = 0.5$.

**Datasets** Our experiments employed the following datasets: (i) The aYahoo dataset (Farhadi et al., 2009b), consisting of 1850 training and 794 test images from 12 classes (e.g., bag, goat, mug). Each image is labeled with 64 binary attributes. The images are collected from Yahoo. (ii) a Pascal dataset (Farhadi et al., 2009b) consisting of 20 classes and 2869 training and 2227 test images. The images are labeled with the same 64 binary attributes as in the aYahoo dataset. (iii) Animals with attributes dataset (AwA2) (Xian et al., 2018) consisting of 26125 training and 11197 test images from 50 animal classes, where each class is labeled with 85 numeric attribute values, (iv) CUB-200-2011 (Wah et al., 2011) consisting of 200 bird species classes, where each image out of 5993 training and 5794 test images is labeled with 312 binary attribute values. We used the standard train/evaluation splits of the datasets.

**Runtime and Infrastructure** The experiments were run on three GeForce RTX 3090 GPUs. Our method completes an epoch every 7 minutes on aYahoo, 10 minutes on aPascal, 1 hour on AwA2, and 1 hour on CUB-200-2011. On average, our method runs for approximately 12 epochs on both aYahoo and aPascal, 2-4 epochs on AwA2, and 35-40 on CUB-200-2011. The exact time withing this range depends on the representation capacity (e.g., number of bits, dimension).

**Baseline methods** We compare our method with various methods that capture a wide variety of approaches: (i) SDT (Frosst and Hinton, 2017) and ANT (Tanno et al., 2019), train a neural network of a tree architecture with the intention of learning high-level concepts. (ii) WS-DAN (Hu et al., 2019) is a method that generates attention maps representing the object's discriminative parts in order to improve the classification. (iii) DFL-CNN and Nts (Wang et al., 2018; Yang et al., 2018) use a network that captures class-specific discriminative regions. Region Grouping (Huang and Li, 2020) is a similar method, which also uses a regularization term that enforces the empirical distribution of part occurrence to align a U-shaped prior distribution. (iv) ProtoPNet (Chen et al., 2019) computes similarity scores of informative patches in the image with learned prototype images. These similarity scores are then aggregated by an MLP classifier. All baselines, except ProtoPNet, are provided with labeled samples, without access to ground-truth semantic segmentation of any kind. ProtoPNet employs cropping-based pre-processing that makes use of the bounding boxes provided with the CUB-200-2011 dataset.

To measure the baselines' and our method's performance on our task, we apply quantization to the penultimate layer of each model. For each method, we apply configurations with penultimate layer dimension $128, 256$ or $512$ and apply uniform quantization with a $1, 2, 3$ or $4$-bit representation. We report, for each method, the results of the configuration that provides the highest accuracy rate on the validation set. Note that this selection is based on accuracy, not on the feature fidelity score. The fidelity score is computed as follows: $d_D(F) := d_D(f; b \circ q \circ F)$, where $F$ is the penultimate layer of the model and $q, b$ are the uniform quantization and binarization operators.

**Quantitative Analysis** In Tab. 1 we report the results of our method and of the various baselines for each dataset. As can be seen, our method obtains a significantly higher feature fidelity score than the baselines across all datasets. We note that ProtoPNet (Chen et al., 2019) makes use of cropping-based preprocessing. In order to compare its results with those of the other methods fairly, we omitted pre-processing on all datasets, except for CUB-200-2011. As can be seen, our method achieves a fidelity score that is higher than ProtoPNet's on CUB-200-2011, even though this kind of supervision helps ProtoPNet achieve the most competitive feature fidelity score among the baselines.

Table 1: **Comparing the performance of various baselines with our method.** We report the classification accuracy rate (Acc) and the feature fidelity score ($d_D(F)$) for each method on each odataset. For ProtoPNet on CUB-200-2011, we report the results with (right) and without (left) using cropping-based augmentations. As can be seen, our method outperforms the other methods in terms of recovering the semantic attributes across datasets.

| | CUB-200-2011 | | | | aYahoo | | | | AwA2 | | | | aPascal | | | |
|---|---|---|---|---|---|---|---|---|---|---|---|---|---|---|---|---|
| Method | Bits | Dim | Acc | $d_D(F)$ | Bits | Dim | Acc | $d_D(F)$ | Bits | Dim | Acc | $d_D(F)$ | Bits | Dim | Acc | $d_D(F)$ |
| SDT | 4 | 512 | 9.80% | 33.01 | 3 | 512 | 17.58% | 31.90 | 2 | 256 | 7.21% | 66.17 | 2 | 128 | 30.48% | 31.53 |
| ANT | 2 | 512 | 11.13% | 33.69 | 4 | 512 | 35.66% | 39.35 | 3 | 256 | 14.17% | 69.27 | 3 | 256 | 31.70% | 39.24 |
| PMG | 4 | 512 | 86.41% | 39.06 | 3 | 512 | 96.14% | 43.68 | 1 | 256 | 96.14% | 71.70 | 2 | 128 | **79.22%** | 45.82 |
| WS-DAN | 4 | 256 | 87.54% | 37.90 | 2 | 256 | 96.82% | 56.90 | 4 | 512 | 96.22% | 78.08 | 3 | 512 | 79.03% | 57.44 |
| DFL-CNN | 3 | 512 | 85.88% | 37.56 | 3 | 512 | 88.16% | 30.06 | 4 | 256 | 89.20% | 65.01 | 4 | 256 | 72.82% | 43.13 |
| Nts | 2 | 128 | 85.42% | 40.66 | 3 | 128 | 98.11% | 55.27 | 4 | 256 | 96.12% | 79.40 | 3 | 128 | 76.10% | 53.42 |
| API-Net | 3 | 512 | **88.21%** | 36.77 | 3 | 512 | **98.72%** | 57.26 | 4 | 256 | **96.25%** | 79.85 | 3 | 256 | 79.15% | 57.58 |
| ProtoPNet | 3 | 512 | 78.82% | 35.31 | 4 | 256 | 85.62% | 49.67 | 2 | 512 | 89.11% | 76.42 | 2 | 512 | 73.52% | 49.66 |
| Region Group | 3 | 512 | 86.10% | 40.45 | 2 | 512 | 97.70% | 58.90 | 3 | 512 | 96.11% | 80.27 | 3 | 256 | 77.27% | 56.80 |
| Quantized network | 4 | 512 | 80.08% | 40.22 | 1 | 512 | 98.08% | 55.75 | 2 | 512 | 95.30% | 80.32 | 2 | 256 | 74.80% | 56.30 |
| Quantized net + DT | 4 | 512 | 39.04% | 32.11 | 1 | 512 | 58.24% | 35.77 | 2 | 512 | 52.09% | 68.77 | 2 | 256 | 47.65% | 33.82 |
| Quantized net + L1 | 4 | 512 | 80.61% | 41.82 | 1 | 512 | 97.57% | 59.20 | 2 | 512 | 94.81% | 83.31 | 2 | 256 | 75.06% | 59.80 |
| Our method | 4 | 512 | 79.82% | **42.58** | 1 | 512 | 96.86% | **61.46** | 2 | 512 | 94.48% | **84.67** | 2 | 256 | 75.09% | **61.72** |

On the other datasets, where no such crops were available, and where ProtoPNet is trained on the entire frame, the feature fidelity score of ProtoPNet is significantly worse than that of our method.

**Ablation Study**   We conducted several ablation studies in order to validate the soundness of our method. Throughout the ablations, we compared our method with three of its variations: (i) a quantized network trained to minimize the cross-entropy loss, (ii) a quantized network trained to minimize the cross-entropy loss and the $L_1$ regularization loss  and (iii) a quantized network with a differentiable soft decision tree (SDT) on top of it, trained to minimize the cross-entropy loss. In all three cases, the neural network is initialized with weights pre-trained on ILSVRC2012 (Russakovsky et al., 2015). We used the online learning decision trees of Domingos and Hulten (2000).  The quantized network uses the same architecture as our model and is trained with the same hyperparameters.

As can be seen in Tab. 1, our method significantly improves the discovery of the semantic attributes, at the small expense of a slight decrease in classification accuracy, which is a secondary priority for our task.

To validate that the obtained performance gap in the fidelity score $d_D(F)$ is consistent across multiple configurations, we report in Fig 2 the results of the quantized neural network baseline method and our complete method.  In Fig 3, we report the obtained classification accuracy per classification. As can be seen, our method does not harm the models' accuracy, while it generally improves the feature fidelity score across the various configurations of feature dimensions and the number of bits used in the binarization process.

A qualitative analysis of the attributes that emerge within the learned representations when our full method is applied is presented in Appendix C.  A short study of the training dynamics of the Intersection Regularization loss, CE loss, accuracy and fidelity score is provided in Appendix D.

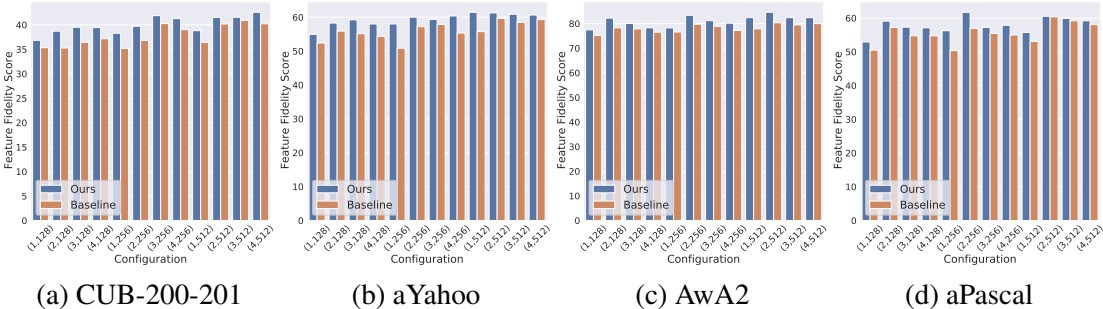

Figure 2: Comparing the performance of our method (blue) and a standard quantized neural network (orange) with respect to the feature fidelity score ($d_D(F)$). The results are shown for different configurations of feature dimensions $n$ and number of bits $k$ for binarization, labeled as $(n, k)$.

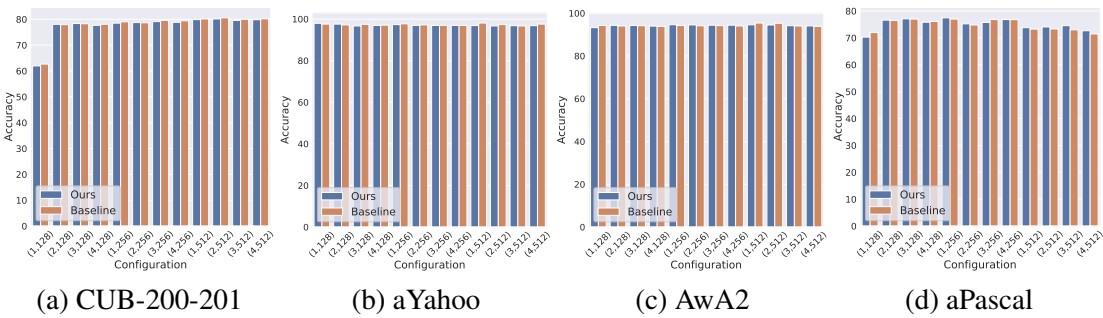

Figure 3: Comparing the performance of our method (blue) and a standard quantized neural network (orange) with respect to classification accuracy rate. The results are shown for each configuration $(n, k)$, where $n$ is the feature dimension and $k$ is the number of quantization bits.

## 7. Limitations

There are a few limitations to both our problem setting and algorithmic approach. First, the problem of "Weakly Supervised Discovery of Semantic Attributes" discussed in Sec. 3 is ill-posed in the general case. This is because we do not have access to the ground-truth binary attributes, and potentially the function $f(x)$ could be replaced with a different function, even when assuming that $y(x)$ is a function of $f(x)$. For example, in general, if $y(x) = \text{sign}(\sum_i f(x)_i)$, then, there are typically multiple functions $f(x)$ that could implement $y(x)$.

In particular, it is typically unknown what the dimension of $F$ ought to be. This is no different from other unsupervised learning settings, such as clustering (Shalev-Shwartz and Ben-David, 2014; Ben-David, 2018), cross-domain mapping (He et al., 2016; Zhu et al., 2017; Kim et al., 2017; Galanti et al., 2018), domain adaptation (Ben-David et al., 2006; Mansour, 2009; Mansour et al., 2009; Ben-David et al., 2010), causality and disentanglement (Peters et al., 2017; Locatello et al., 2019) and sparse dictionary learning (Hillar and Sommer, 2015; Garfinkle and Hillar, 2019), where ambiguity is an inherent aspect of the learning problem.

This issue makes this problem challenging, especially when the number of ground-truth attributes is large, as in the case of CUB-200-2011, or when there are multiple redundant attributes in the dataset.

To cope with the multiple-admissible-solution issue in unsupervised learning, one typically make assumptions on the structure of the target function, which narrows the space of functions captured by the algorithm. In our case, we assume that $y(x)$ can be represented as $t(f(x))$, where $t$ is a decision tree of small depth.

## 8. Conclusions

In this paper, we introduced the problem of weakly supervised discovery of semantic attributes. This problem explores the emergence of interpretable features quantitatively, based on a set of semantic features that is unseen during training. We present evidence that methods which are tailored to extract semantic attributes do not necessarily perform well in this metric and suggest a new approach to solving this problem.

Our method is based on joint learning in the soft intersection of two hypothesis classes: a neural network for obtaining its classification power, and a decision tree for learning features that lend themselves to classification by short logical expressions. We demonstrate that this way of learning is more effective than other methods for the new task and that learning with the new regularization scheme improves the fidelity of the obtained feature map.

## Acknowledgments

This project has received funding from the European Research Council (ERC) under the European Unions Horizon 2020 research and innovation programme (grant ERC CoG 725974).

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

## Appendix A. Proofs of Props. 1 and 2

**Proposition 1** *Assume that $\mathcal{Q}(\theta, \omega)$ is convex and $\beta$-smooth w.r.t $\theta$ for any fixed value of $\omega$. Let $\theta_1$ be an initialization and $\omega_1 \in \arg\min_\omega \mathcal{Q}(\theta_1, \omega)$. We define $\theta_t$ to be the weights produced after $t$ iterations of applying Gradient Descent on $\mathcal{Q}(\theta, \omega_{t-1})$ over $\theta$ with learning rate $\mu < \beta^{-1}$ and $\omega_t = \arg\min_\omega \mathcal{Q}(\theta_{t-1}, \omega)$. Then, we have*

$$\lim_{t\to\infty} \mathcal{Q}(\theta_t, \omega_t) = \lim_{t\to\infty} \min_\theta \mathcal{Q}(\theta, \omega_t) = \lim_{t\to\infty} \min_\omega \mathcal{Q}(\theta_t, \omega).$$

**Proof** First, we would like to prove that $\mathcal{Q}(\theta_t, \omega_t)$ is monotonically decreasing. We notice that for each $t \in \mathbb{N}$, we have

$$\mathcal{Q}(\theta_t, \omega_t) \le \mathcal{Q}(\theta_t, \omega_{t-1}) \tag{7}$$

since $\omega_t$ is the global minimizer of $\mathcal{Q}(\theta_t, \omega)$. In addition, by the proof of (cf. Nesterov (2014), Thm. 2.1.14), we have

$$\mathcal{Q}(\theta_t, \omega_{t-1}) \le \mathcal{Q}(\theta_{t-1}, \omega_{t-1}) - \eta \|\nabla_\theta \mathcal{Q}(\theta_t, \omega_{t-1})\|_2^2 \le \mathcal{Q}(\theta_{t-1}, \omega_{t-1}), \tag{8}$$

where $\eta := \mu(1 - 0.5\beta\mu) > 0$. Therefore, we conclude that $\mathcal{Q}(\theta_t, \omega_t)$ is indeed monotonically decreasing. Since $\mathcal{Q}$ is non-negative, we conclude that $\mathcal{Q}(\theta_t, \omega_t)$ is a convergent sequence. In particular, $\theta_t$ is bounded. Otherwise, $\mathcal{Q}(\theta_t, \omega_t) \geq \mathcal{L}_{\mathcal{S}}[h_{\theta_t}, y] \to \infty$ in contradiction to the fact that $\mathcal{Q}(\theta_t, \omega_t)$ is a convergent sequence.

Next, by applying the convexity of $\mathcal{Q}(\theta, \omega_{t-1})$ (as a function of $\theta$), we have

$$\mathcal{Q}(\theta_t, \omega_{t-1}) - \mathcal{Q}(\theta_t^*, \omega_{t-1}) \leq \langle \nabla_\theta \mathcal{Q}(\theta, \omega_{t-1}), \theta_t - \theta_t^* \rangle \leq \|\nabla_\theta \mathcal{Q}(\theta_t, \omega_{t-1})\|_2 \cdot \|\theta_t - \theta_t^*\|_2, \quad (9)$$

where $\theta_t^* = \arg\min_\theta \mathcal{Q}(\theta, \omega_t)$. By combining Eqs. (8) and (9), we have

$$\mathcal{Q}(\theta_t, \omega_t) \leq \mathcal{Q}(\theta_t, \omega_{t-1}) \leq \mathcal{Q}(\theta_{t-1}, \omega_{t-1}) - \frac{\eta}{\|\theta_t - \theta_t^*\|_2^2} \left( \mathcal{Q}(\theta_{t-1}, \omega_{t-1}) - \min_\theta \mathcal{Q}(\theta, \omega_{t-1}) \right)^2 \tag{10}$$

In particular,

$$\mathcal{Q}(\theta_{t-1}, \omega_{t-1}) - \mathcal{Q}(\theta_t, \omega_t) \geq \frac{\eta}{\|\theta_t - \theta_t^*\|_2^2} \left( \mathcal{Q}(\theta_{t-1}, \omega_{t-1}) - \min_\theta \mathcal{Q}(\theta, \omega_{t-1}) \right)^2 \tag{11}$$

Since the left-hand side tends to zero and the right-hand side is lower bounded by zero, by the sandwich theorem the right-hand side tends to zero as well. Since both $\theta_t$ and $\theta_t^*$ are bounded sequences ($\{\arg\min_\theta \mathcal{Q}(\theta, \omega) \mid \omega \in \Omega\}$ is well-defined and bounded), we conclude that $\lim_{t\to\infty} \mathcal{Q}(\theta_t, \omega_t) = \lim_{t\to\infty} \min_\theta \mathcal{Q}(\theta, \omega_t)$. We also have: $\mathcal{Q}(\theta_t, \omega_t) = \min_\omega \mathcal{Q}(\theta_t, \omega)$ by definition, and therefore, $\lim_{t\to\infty} \mathcal{Q}(\theta_t, \omega_t) = \lim_{t\to\infty} \min_\omega \mathcal{Q}(\theta_t, \omega)$ as well. ∎

**Proposition 2** *Assume that $\mathcal{Q}(\theta, \omega)$ is a twice continuously differentiable, element-wise convex (i.e., convex w.r.t $\theta$ for any fixed value of $\omega$ and vice versa), Lipschitz continuous and $\beta$-smooth function, whose saddle points are strict. Let $\theta_t, \omega_t$ be the weights produced after $t$ iterations of applying BCGD on $\mathcal{Q}(\theta, \omega)$ with learning rate $\mu < \beta^{-1}$. $(\theta_t, \omega_t)$ then converges to a local minimum $(\hat{\theta}, \hat{\omega})$ of $\mathcal{Q}$ that is also an equilibrium point.*

**Proof** Firstly, since $\mathcal{Q}(\theta, \omega)$ is a twice continuously differentiable, Lipschitz continuous and $\beta$-smooth function, by Prop. 3.4 and Cor. 3.1 in (Song et al., 2017), $(\theta_t, \omega_t)$ converge to a local minimum $(\hat{\theta}, \hat{\omega})$ of $\mathcal{Q}$. Therefore, it is left to show that $(\hat{\theta}, \hat{\omega})$ is also an equilibrium point. We note that $\mathcal{Q}(\theta, \omega)$ is element-wise convex and $\beta$-smooth. By the proof of (cf. Nesterov (2014), Thm. 2.1.14), we have

$$\mathcal{Q}(\theta_{t+1}, \omega_t) \leq \mathcal{Q}(\theta_t, \omega_t) - \eta\|\nabla_\theta \mathcal{Q}(\theta_t, \omega_t)\|_2^2, \tag{12}$$

where $\eta := \mu(1 - 0.5\beta\mu) > 0$. By applying the convexity of $\mathcal{Q}(\theta, \omega_t)$ (as a function of $\theta$), we have

$$\begin{aligned} \mathcal{Q}(\theta_t, \omega_t) - \mathcal{Q}(\theta_t^*, \omega_t) &\leq \langle \nabla_\theta \mathcal{Q}(\theta_t, \omega_t), \theta_t - \theta_t^* \rangle \\ &\leq \|\nabla_\theta \mathcal{Q}(\theta_t, \omega_t)\|_2 \cdot \|\theta_t - \theta_t^*\|_2, \end{aligned} \tag{13}$$

where $\theta_t^* = \arg\min_\theta \mathcal{Q}(\theta, \omega_t)$ and $\omega_t^* = \arg\min_\omega \mathcal{Q}(\theta_{t+1}, \omega)$. By combining Eqs. (12) and (13), we have

$$\mathcal{Q}(\theta_{t+1}, \omega_t) \leq \mathcal{Q}(\theta_t, \omega_t) - \frac{\eta}{\|\theta_t - \theta_t^*\|_2^2} \left( \mathcal{Q}(\theta_t, \omega_t) - \min_\theta \mathcal{Q}(\theta, \omega_t) \right)^2, \tag{14}$$

and similarly, we have

$$\mathcal{Q}(\theta_{t+1}, \omega_{t+1}) \leq \mathcal{Q}(\theta_{t+1}, \omega_t) - \frac{\eta}{\|\omega_t - \omega_t^*\|_2^2} \left( \mathcal{Q}(\theta_{t+1}, \omega_t) - \min_\omega \mathcal{Q}(\theta_{t+1}, \omega) \right)^2. \quad (15)$$

In particular, we have

$$\begin{aligned}
\mathcal{Q}(\theta_{t+1}, \omega_{t+1}) \leq &\mathcal{Q}(\theta_t, \omega_t) - \frac{\eta}{\|\theta_t - \theta_t^*\|_2^2} \left( \mathcal{Q}(\theta_t, \omega_t) - \min_\theta \mathcal{Q}(\theta, \omega_t) \right)^2 \\
&- \frac{\eta}{\|\omega_t - \omega_t^*\|_2^2} \left( \mathcal{Q}(\theta_{t+1}, \omega_t) - \min_\omega \mathcal{Q}(\theta_{t+1}, \omega) \right)^2.
\end{aligned} \quad (16)$$

Therefore, the sequence $\mathcal{Q}(\theta_t, \omega_t)$ is monotonically decreasing. Since $\mathcal{Q}(\theta_t, \omega_t)$ is non-negative, it converges to some non-negative constant. By Eq. (16), we have

$$\begin{aligned}
&\mathcal{Q}(\theta_{t+1}, \omega_{t+1}) - \mathcal{Q}(\theta_t, \omega_t) \\
\geq &\frac{\eta}{\|\theta_t - \theta_t^*\|_2^2} \left( \mathcal{Q}(\theta_t, \omega_t) - \min_\theta \mathcal{Q}(\theta, \omega_t) \right)^2 + \frac{\eta}{\|\omega_t - \omega_t^*\|_2^2} \left( \mathcal{Q}(\theta_{t+1}, \omega_t) - \min_\omega \mathcal{Q}(\theta_{t+1}, \omega) \right)^2.
\end{aligned} \quad (17)$$

Since the left-hand side tends to zero and the right-hand side is lower bounded by zero, by the sandwich theorem the sequences

$$\frac{\eta}{\|\theta_t - \theta_t^*\|_2^2} \left( \mathcal{Q}(\theta_t, \omega_t) - \min_\theta \mathcal{Q}(\theta, \omega_t) \right)^2$$

and

$$\frac{\eta}{\|\omega_t - \omega_t^*\|_2^2} \left( \mathcal{Q}(\theta_{t+1}, \omega_t) - \min_\omega \mathcal{Q}(\theta_{t+1}, \omega) \right)^2$$

tend to zero. We note that $\theta_t$ and $\omega_t$ are convergent sequences and are, therefore, also bounded. In addition, we recall that the sets $\{\arg\min_\theta \mathcal{Q}(\theta, \omega) \mid \omega \in \Omega\}$ and $\{\arg\min_\omega \mathcal{Q}(\theta, \omega) \mid \theta \in \Theta\}$ are well-defined and bounded. Hence, the terms $\|\theta_t - \theta_t^*\|_2^2$ and $\|\omega_t - \omega_t^*\|_2^2$ are bounded. Thus, we conclude that the sequences $(\mathcal{Q}(\theta_t, \omega_t) - \min_\theta \mathcal{Q}(\theta, \omega_t))^2$ and $(\mathcal{Q}(\theta_{t+1}, \omega_t) - \min_\omega \mathcal{Q}(\theta_{t+1}, \omega))^2$ tend to zero. In particular,

$$\begin{aligned}
\lim_{t\to\infty} \mathcal{Q}(\theta_t, \omega_t) &= \lim_{t\to\infty} \min_\theta \mathcal{Q}(\theta, \omega_t) \\
\lim_{t\to\infty} \mathcal{Q}(\theta_{t+1}, \omega_t) &= \lim_{t\to\infty} \min_\omega \mathcal{Q}(\theta_t, \omega)
\end{aligned} \quad (18)$$

Since $\mathcal{Q}$ is a Lipschitz continuous function, we have (i) $\lim_{t\to\infty} \mathcal{Q}(\theta_{t+1}, \omega_t) = \lim_{t\to\infty} \mathcal{Q}(\theta_t, \omega_t) = \mathcal{Q}(\hat{\theta}, \hat{\omega})$, (ii) $\lim_{t\to\infty} \min_\theta \mathcal{Q}(\theta, \omega_t) = \min_\theta \mathcal{Q}(\theta, \hat{\omega})$ and (iii) $\lim_{t\to\infty} \min_\omega \mathcal{Q}(\theta_t, \omega) = \min_\omega \mathcal{Q}(\hat{\theta}, \omega)$. Therefore, we finally conclude that $(\hat{\theta}, \hat{\omega})$ is an equilibrium point of $\mathcal{Q}$. $\blacksquare$

## Appendix B. Uniform Quantization

The uniform quantization method is fully described in Alg. 2. For a given function $p$, we denote by $\partial p$ the estimated gradients of $p$ w.r.t $x$. In Alg. 2, the estimated gradient of $\tilde{z}$ is computed as the gradient of $\min(\max(z_{init}, q_{min}), q_{max})$ (since $\partial \text{round}(x) = 1$ when applying STE Bengio et al. (2013b)), which is simply $\mathbb{1}[z_{init} \in (q_{min}, q_{max})] \cdot \frac{\partial z_{init}}{\partial x}$, where $\frac{\partial z_{init}}{\partial x}$ is the gradient of $z_{init}$ w.r.t $x$. $\partial \tilde{q}_i$ and $\partial q$ are defined similarly, see lines 10-11 in Alg. 2.

---

**Algorithm 2** The uniform quantization method (forward and backward passes)

---

**Require:** $x$ is tensor to be quantized; $r$ number of bits.

1: $q_{\min}, q_{\max} = 0, 2^r - 1$;
2: $x_{\min}, x_{\max} = \min_j\{x_j\}, \max_j\{x_j\}$;
3: $s = \frac{x_{\max} - x_{\min}}{q_{\max} - q_{\min}}$;
4: $z_{\text{init}} = \frac{q_{\min} - x_{\min}}{s}$;
5: $z = \min(\max(z_{\text{init}}, q_{\min}), q_{\max})$;
6: $\tilde{z} = \text{round}(z)$;
7: $\forall i : \ \tilde{q}_i = \tilde{z} + \frac{x_i}{s}$;
8: $\forall i : \ q_i = \text{round}(\min(\max(\tilde{q}_i, q_{\min}), q_{\max}))$;
9: $\partial \tilde{z} = \mathbb{1}[z_{init} \in (q_{\min}, q_{\max})] \cdot \frac{\partial z_{init}}{\partial x}$;
10: $\forall i : \ \partial \tilde{q}_i = \partial \tilde{z} + \frac{\partial (x_i/s)}{\partial x}$;
11: $\forall i : \ \partial q_i = \mathbb{1}[\tilde{q}_i \in (q_{\min}, q_{\max})] \cdot \partial \tilde{q}_i$;
12: **return** $q, \partial q$;

---

## Appendix C.  Qualitative Analysis

We provide a qualitative analysis of the advantage of our method over a simple neural network with a quantized representation layer. Each method learns a binarized feature map $b \circ F$. In this experiment, for each learned feature map $F$ and a given ground-truth attribute $q_i(x)$, we compute $d_D(q_i; b \circ q \circ F)$ that measures whether it appears in the learned binarized representation layer $b \circ q \circ F$. In Figs. 4-6 we plot the values of $d_D(q_i; b \circ q \circ F)$ for each attribute in the dataset as a heatmap. We also report the number of attributes for which our method achieves a higher score (by at least $\epsilon = 0.05$) and vice versa.

| 2D Boxy | 3D Boxy | Round | Vert Cyl | Horiz Cyl | Tail | Occluded | Beak |
|---|---|---|---|---|---|---|---|
| Head | Ear | Snout | Nose | Mouth | Hair | Face | Eye |
| Hand | Leg | Foot | Wing | Propeller | Jet engine | Window | Row Wind |
| Door | Headlight | Taillight | Side Mirror | Pedal | Handlebars | Sail | Mast |
| Leaf | Flower | Stem | Wool | Clear | Skin | Metal | Plastic |
| Shiny | Vegetation | Wood | Cloth | Hern | Rein | Exhaust | Trunk |
| Exhaust | Saddle | Shiny | Furn. Leg | Furn. Back | Furn. Seat | Furn. Arm | Glass |
| Screen | Arm | Torso | Wheel | Label | Leather | Jen Engine | Pot |

| 2D Boxy | 3D Boxy | Round | Vert Cyl | Horiz Cyl | Tail | Occluded | Beak |
|---|---|---|---|---|---|---|---|
| Head | Ear | Snout | Nose | Mouth | Hair | Face | Eye |
| Hand | Leg | Foot | Wing | Propeller | Jet engine | Window | Row Wind |
| Door | Headlight | Taillight | Side Mirror | Pedal | Handlebars | Sail | Mast |
| Leaf | Flower | Stem | Wool | Clear | Skin | Metal | Plastic |
| Shiny | Vegetation | Wood | Cloth | Hern | Rein | Exhaust | Trunk |
| Exhaust | Saddle | Shiny | Furn. Leg | Furn. Back | Furn. Seat | Furn. Arm | Glass |
| Screen | Arm | Torso | Wheel | Label | Leather | Jen Engine | Pot |

Figure 4: **Heatmaps of** $d_D(q_i; b \circ q \circ F)$ **on the aYahoo dataset (**Farhadi et al., 2009b**).** In **(top)** we report the results of our method. **(bottom)** reports the results of a standard neural network with a quantized representation layer. Our method obtains a higher score on 23 attributes and a lower score on 4 attributes.

| 2D Boxy | 3D Boxy | Round | Vert Cyl | Horiz Cyl | Tail | Occluded | Beak |
|---|---|---|---|---|---|---|---|
| Head | Ear | Snout | Nose | Mouth | Hair | Face | Eye |
| Hand | Leg | Foot | Wing | Propeller | Jet engine | Window | Row Wind |
| Door | Headlight | Taillight | Side Mirror | Pedal | Handlebars | Sail | Mast |
| Leaf | Flower | Stem | Wool | Clear | Skin | Metal | Plastic |
| Shiny | Vegetation | Wood | Cloth | Hern | Rein | Exhaust | Trunk |
| Exhaust | Saddle | Shiny | Furn. Leg | Furn. Back | Furn. Seat | Furn. Arm | Glass |
| Screen | Arm | Torso | Wheel | Label | Leather | Jen Engine | Pot |

| 2D Boxy | 3D Boxy | Round | Vert Cyl | Horiz Cyl | Tail | Occluded | Beak |
|---|---|---|---|---|---|---|---|
| Head | Ear | Snout | Nose | Mouth | Hair | Face | Eye |
| Hand | Leg | Foot | Wing | Propeller | Jet engine | Window | Row Wind |
| Door | Headlight | Taillight | Side Mirror | Pedal | Handlebars | Sail | Mast |
| Leaf | Flower | Stem | Wool | Clear | Skin | Metal | Plastic |
| Shiny | Vegetation | Wood | Cloth | Hern | Rein | Exhaust | Trunk |
| Exhaust | Saddle | Shiny | Furn. Leg | Furn. Back | Furn. Seat | Furn. Arm | Glass |
| Screen | Arm | Torso | Wheel | Label | Leather | Jen Engine | Pot |

Figure 5: **Heatmaps of** $d_D(q_i; b \circ q \circ F)$ **on the aPascal dataset (**Farhadi et al., 2009b**).** In **(top)** we report the results of our method. **(bottom)** reports the results of a standard neural network with a quantized representation layer. Our method obtains a higher score on 38 attributes and a lower score on 3 attributes.

Figure 6: **Heatmaps of** $d_D(q_i; b \circ q \circ F)$ **on the AwA2 dataset (Xian et al., 2018).** In **(top)** we report the results of our method. **(bottom)** reports the results of a standard neural network with a quantized representation layer. Our method obtains a higher score on 48 attributes and a lower score on 3 attributes.

## Appendix D. Training Dynamics

In the following, we show the training dynamics over aPascal and aYahoo datasets. As can be seen, as a general tendency, the fidelity score increases in correlation with the intersection regularization loss (IR Loss) decreases. While during most of the training process both metrics improve together, at later stages they seem to fluctuate in a narrow band in a matter that is not always correlated. The cross-entropy (CE) loss and the accuracy constantly improve from epoch to epoch. Therefore, we do not see a clear tradeoff between the IR loss and the CE loss, at least in the first 12 epochs. =

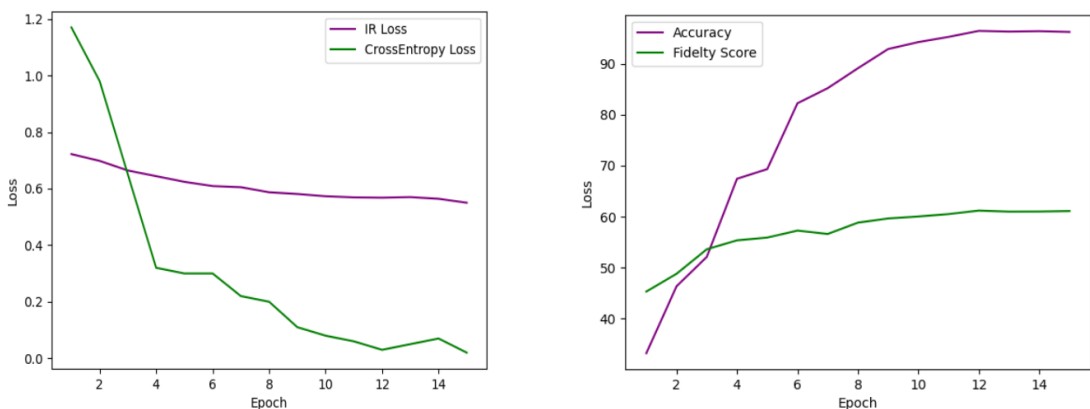

Figure 7: **The training dynamics over aYahoo dataset.**

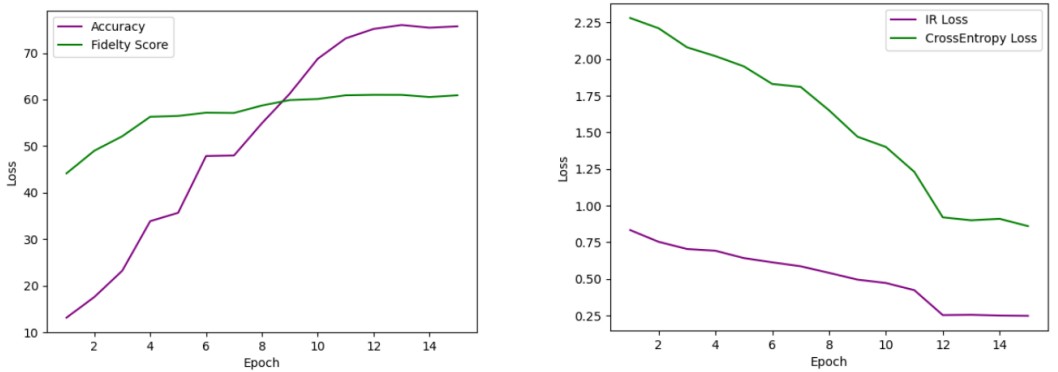

Figure 8: **The training dynamics over aPascal dataset.**

