# OpenReview forum: "Weakly Supervised Discovery of Semantic Attributes"
_cclear.cc/CLeaR/2022/Conference — CLeaR 2022 Poster_

### Official Review · Reviewer_jvUt · 2021-11-18

**Confidence:** 3
**Overall Score:** 6

**Main Review:**

Strengths:
The results appear promising: according to the newly defined measure of feature fidelity, the proposed model learns more interpretable discrete representations without much loss on the performance.

The idea of hypothesis-space regularization is interesting (but it could be better motivated, see below).

The set of baselines and datasets makes sense.

Weaknesses:
It is not clear that the paper fits well with the causal learning and reasoning theme. While learning discrete representations is likely to be useful in the field of causal ML, the paper does not discuss these connections.

The paper does not motivate the intuition behind the hypothesis-regularization idea. Why should we expect it to work? How the proposed training dynamic is expected to enforce feature fidelity in the discrete representation.

An important aspect of the approach is the training dynamic arising from alternating between updates on the two losses and the tree retraining. But the experiments do not show this dynamic, they only focus on the final outcome.

The number of dimensions in the discrete layer seems to be chosen much larger than the annotations used for evaluation (256, 512 in networks versus 64, 50 in datasets). How does the proposed model behave when the number of dimensions is the same or smaller than in the annotations? In Figure 3. some ablation is performed but the number of dimensions still remains >= 128.



Minor comments:
In the computation of feature fidelity: what strategy is used to compare 4-bits attributes with binary attributes, or to compare with numeric attributes?

In section 3.1
max{r(q1, q2;D), r(q1, 1 − q2;D)} shouldn't it be 1 - q1 as well in the second term to have invariance w.r.t. whether the positive are marked as 1 or 0.

**Summary:**

The paper proposes to learn an interpretable discrete layer with both quantization and regularization with decision trees.

---

> ### Author Response · Authors · 2021-12-03
> **Thank you for your review (part 1/2)**
>
> Thank you for your supportive review, we appreciate the time and effort.
>
> > **Reviewer:** It is not clear that the paper fits well with the causal learning and reasoning theme. While learning discrete representations is likely to be useful in the field of causal ML, the paper does not discuss these connections.
>
> **Authors:** The paper fits well within the topics listed in the call for papers under “Fairness, accountability, transparency, explainability, trustworthiness, and recourse” and the links to transparency and explainability are made within the manuscript (e.g., Sections 1 and 2). More broadly, we believe that the discovery of meaningful intermediate features is at the heart of the scientific methodology and is therefore crucial for the study of reasoning.
>
> > **Reviewer:** The paper does not motivate the intuition behind the hypothesis-regularization idea. Why should we expect it to work? How the proposed training dynamic is expected to enforce feature fidelity in the discrete representation.
>
> **Answer:** Motivation: As mentioned in the introduction, we look for features that are both evidence-based and distinctive. By evidence-based we mean that there is a mapping $f$ such that every input $x$ is mapped to a vector $f(x)$ of binary values, indicating the presence of each attribute. Distinctiveness means that there should be simple rules that determine the class label $y(x)$ based on the obtained attributes $f(x)$. Since deep networks excel in the creating mappings such as $f$ and decision trees are designed to create logical rules that recover labels from attributes, finding a classifier that is in the intersection of both hypothesis classes (the one of deep networks and the one of decision trees) seem like a plausible way to obtain the desired properties.
>
> We prove, in Sec. 4 that the proposed method for intersecting the hypothesis spaces converges under some assumptions.
>
> For dynamics, please see below.
>
> > **Reviewer:** An important aspect of the approach is the training dynamic arising from alternating between updates on the two losses and the tree retraining. But the experiments do not show this dynamic, they only focus on the final outcome.
>
> **Authors:** Following the reviews, we provide plots of the dynamics of the IR loss and the fidelity score over training epochs for various datasets.
>
> aPascal dataset: https://imgur.com/a/hVPOZ6J  (anonymous image)
>
> aYahoo dataset : https://imgur.com/a/LPID9Cm (anonymous image)
>
> As can be seen, as a general tendency, the fidelity score increases in correlation with the fact that the intersection regularization loss (IR Loss) decreases. While during most of the training process both metrics improve together, at later stages they seem to fluctuate in a narrow band in a matter that is not always correlated. The cross-entropy (CE) loss and the accuracy constantly improve from epoch to epoch. There does not seem to be a clear tradeoff between the IR loss and the CE loss, at least in the first 12 epochs.

---

> > ### Author Response · Authors · 2021-12-03
> > **Thank you for your review (part 2/2)**
> >
> > > **Reviewer:** The number of dimensions in the discrete layer seems to be chosen much larger than the annotations used for evaluation (256, 512 in networks versus 64, 50 in datasets). How does the proposed model behave when the number of dimensions is the same or smaller than in the annotations? In Figure 3. some ablation is performed but the number of dimensions still remains $\geq 128$.
> >
> > **Authors:** For most datasets, the number of ground-truth attributes in the datasets is between 60-85 (except for CUB-200-2011, where we have 312 attributes), and thus lowering the latent dimension to less than 100 might lead to an excessive loss in expressivity, which would decrease both the accuracy and the fidelity score.
> >
> > In our experiments, we chose the dimension to be the minimal dimension (that is a power of 2) that allows a reasonable accuracy rate. This way we can ensure that the model is able to perform reasonably on the classification task, while using a relatively small number of dimensions for recovering the attributes.
> >
> > With the default parameters of the paper (see also parameter sensitivity study in Sec. 7), we obtain for aYahoo an accuracy of 96.86%, and fidelity score of: 61.46.
> >
> > For latent dimension of 64 and quantization with 1 bit both the accuracy and the fidelity are hurt: accuracy: 92.22%, and fidelity score of: 48.18.
> >
> > Reducing the dimension further, this becomes worse: For latent dimension of 32 and quantization with 1 bit we get: accuracy: 91.10%, and fidelity score of: 51.36. For latent dimension of 16 and quantization with 1 bit we get: accuracy: 58.47%, and fidelity score of: 35.72.
> >
> > In our method, when the accuracy is low, the distinctiveness (accuracy obtained with simple rules over the attributes; see introduction) is low, since the two are tied through intersection regularization. Therefore, assuming that one can obtain high accuracy based on the ground truth features, the fidelity (agreement with the ground truth features) is also expected to be low, when the accuracy is low. In conclusion, the accuracy must be high enough in order to support high fidelity, requiring the deep neural network to have enough capacity.
> >
> > > **Reviewer:** In the computation of feature fidelity: what strategy is used to compare 4-bits attributes with binary attributes, or to compare with numeric attributes?
> >
> > **Authors:** The strategy is described in the last paragraph of ‘Quantization and Binarization’ paragraph of Section 5 (on page 8). For multi-bit quantization, each value of each attribute is considered a separate (binary) feature, e.g., for 2-bit quantization possible values of feature $f_i$ are 0,1,2,3; this generates 4 positive boolean (binary) features $f_i=0$, $f_i=1$, $f_i=2$, and $f_i=3$, and four negative ones: $f_i\neq 0$, $f_i\neq 1$, $f_i\neq 2$, and $f_i\neq 3$. For instance, in the 2-bit quantization, the binary attribute $f_i=0$ is encoded as (1,0,0,0), the attribute $f_i=1$ is encoded (0,1,0,0) and the binary attribute $f_i=3$ is encoded as (0,0,0,1).
> >
> > > **Reviewer:** In section 3.1 $\max(r(q_1, q_2;D), r(q_1, 1 − q_2;D))$ shouldn't it be $1 - q_1$ as well in the second term to have invariance w.r.t. whether the positive are marked as 1 or 0.
> >
> > **Authors:** The current expression is correct. Intuitively, we want to compare $q_1$ with $q_2$ and also with $q_2$ as if its “bits were flipped”, $(1-q_2)$. This is given by $\max(r(q_1, q_2;D), r(q_1, 1 − q_2;D))$. For instance, suppose $q_2$ is a function that classifies dogs as 1 and cats as 0, and $q_1$ is the same function but flipped (i.e., classifies dogs as 0 and cats as 1). In this case, $r(q_1, q_2; D)=0$ and $r(1-q_1, 1-q_2; D)=0$, even though the functions $q_1$ and $q_2$ are essentially the same thing (up to bit flip). This matches the following equation $r(q_1, 1-q_2;D)=r(q_1,q_1;D)=1$.

---

### Official Review · Reviewer_uthu · 2021-11-22

**Confidence:** 4
**Overall Score:** 6

**Main Review:**

This paper proposes a novel problem named weakly supervised discovery of semantic attributes, which aims at extracting the semantic attributes based on simply image-level labels. To address this issue, this paper proposes to take advantage of MLP and decision tree to optimize the main classification network. Besides, Intersection Regularization is also proposed for training the MLP the decision tree together, and some relevant empirical results and theoretical analysis are also provided.

Pros:
1.	The paper makes a concrete and comprehensive overview of relevant work, especially in the field of Interpretability.
2.	The paper proposes an interesting method to solve the proposed issue. The MLP behind the feature map is proposed to represent the quantized function for semantic attributes. The decision tree is proposed to auxiliary the main classification network to learn the relationship from attributes to image-level labels. The idea is easily implemented, and such modeling for these two different signals (semantic attributes and classification labels) is reasonable.
3.	This paper also proposes a method named Intersection Regularization to optimize the main classification network by interactively updating the MLP and decision tree, which is also proved to be effective by some experiment results.

Cons:
1.	There are some detailed errors in writing. For instance, page 5 mentioned “i’th elementary vector” and “ith attributes” at the same time. Besides, some nouns and sentences need to be clarified and defined so as to eliminate the ambiguity in this paper. For example, “feature fidelity” in Section 3.1, ”GD” in Section 5, and why “U = R, [-1,1] or {±1}”.
2.	There is a confusing definition to the newly proposed problem. The title is “Weakly supervised Discovery of Semantic Attributes”, but page 2 also defines it as “Weakly Unsupervised Discovery of Semantic Attributes”. Therefore, I suggest the author should make an accurate name or definition to the proposed problem.
3.	The motivation of the proposed problem seems not understandable to me. The essence of weakly supervised/unsupervised learning is to liberate humans from exhaustive annotation work, which means that the network could work with fewer or even no labels. Although this paper aims to find the attributes without annotating them, the image-level labels are utilized during the training process. It may sound more reasonable to utilize noisy or fewer labels for semantic attributes during the training phase so that the problem could be defined as “weakly supervised discovery of attributes”.
4.	The meaning of exploring the potential semantic attributes should serve to better classification. However, the experiment results show that the classification accuracy of proposed method is way lower than some baseline methods, it may not convincing that only the evaluation to the recovered attributes achieves higher performance. Besides, the sole evaluation metric is feature fidelity score, which is newly proposed in this paper. Therefore, the author should show more solid and fair experiment results.
5.	Lack of detailed implementation about the experiments including but not limited to the relevant data augmentation and training hyper-parameters such as batch size.

**Summary:**

This paper proposes a novel problem named weakly supervised discovery of semantic attributes, which aims at extracting the semantic attributes based on simply image-level labels. To address this issue, this paper proposes to take advantage of MLP and decision tree to optimize the main classification network. Besides, Intersection Regularization is also proposed for training the MLP the decision tree together, and some relevant empirical results and theoretical analysis are also provided.

---

> ### Author Response · Authors · 2021-12-03
> **Thank you for your review (part 1/2)**
>
> Thank you for your review, we appreciate the time and effort.
>
> > **Reviewer:** 1. There are some detailed errors in writing. For instance, page 5 mentioned “i’th elementary vector” and “ith attributes” at the same time. Besides, some nouns and sentences need to be clarified and defined so as to eliminate the ambiguity in this paper. For example, “feature fidelity” in Section 3.1, ”GD” in Section 5, and why “U = R, [-1,1] or {±1}”.
>
> **Authors:** Following the reviews, we fixed all of the mentioned typos. Feature fidelity stands for the metric proposed in Section 3.1. We refer to $d_D(f;g)$ as the fidelity of the feature $f$ with respect to $g$. By GD we meant Gradient Descent. Indeed, the setting should not be restricted to $U=\mathbb{R}$, $[-1,1]$ and $\{\pm 1\}$. Following the review, we removed this comment from the paper.
>
> > **Reviewer:** 2. There is a confusing definition to the newly proposed problem. The title is “Weakly supervised Discovery of Semantic Attributes”, but page 2 also defines it as “Weakly Unsupervised Discovery of Semantic Attributes”. Therefore, I suggest the author should make an accurate name or definition to the proposed problem.
>
> **Authors:** The term should be “Weakly Supervised Discovery of Semantic Attributes”. The other term is a typo, which we fixed following the reviews.
>
> > **Reviewer:** 3. The motivation of the proposed problem seems not understandable to me. The essence of weakly supervised/unsupervised learning is to liberate humans from exhaustive annotation work, which means that the network could work with fewer or even no labels. Although this paper aims to find the attributes without annotating them, the image-level labels are utilized during the training process. It may sound more reasonable to utilize noisy or fewer labels for semantic attributes during the training phase so that the problem could be defined as “weakly supervised discovery of attributes”.
>
> **Authors:** We agree that in many weakly/unsupervised learning settings the goal is to predict the labels in scenarios in which they are not present or partially present. As discussed in Section 3, in the proposed setting, the learning algorithm is provided with the class-labels (e.g., cat/ dog/ airplane/ etc) but is not provided with the semantic features $f(x)$ that are associated with pairs of samples $(x,y(x))$, where $x$ is an example and $y$ is its class-label. The goal is to be able to recover the semantic features $f(x)$, which are hidden from the learning algorithm. In that sense, the actual labels that we would like to predict is $f(x)$, rather than $y(x)$. In other words, $y(x)$ serves as auxiliary information, rather than the actual label in this setting. Therefore, we refer to this setting as a weakly supervised recovery of the attributes $f(x)$, since they are not provided to the learning algorithm, but some auxiliary labels (i.e., $y(x)$) that are related to $f(x)$ are provided instead.
>
> This scenario is very common: in almost all scholarly fields there are scenarios in which one has labels but needs to find out the separating features. In literature, one knows the author but not the characteristics of their writings. In chemistry, one knows the melting temperature of the molecule but not the sub-molecules that lead to this value. Section 1 provides other examples in paleography, botany, and zoology.

---

> > ### Author Response · Authors · 2021-12-03
> > **Thank you for your review (part 2/2)**
> >
> > > **Reviewer:** 4a. The meaning of exploring the potential semantic attributes should serve to better classification. However, the experiment results show that the classification accuracy of the proposed method is way lower than some baseline methods...
> >
> > **Authors:** It is often the case that adding explainability hurts overall performance [1]. However, in our case, in a direct comparison between the same generic architecture with and without Intersection Regularization (i.e., ‘quantized net’ and ‘quantized net + L1’), the hit in performance is modest.
> >
> > While some baselines in the literature apply very specialized losses (Zhuang et al. 2020 , Chen et al , 2019) or augmentations (T. Hu et al. 2019) in order to drive accuracy in specific datasets, we decided to avoid these in favor of a more generic approach.
> > [1] G. K. Dziugaite, S. B. David, D. Roy, Enforcing Interpretability and its Statistical Impacts: Trade-offs between Accuracy and Interpretability, Arxiv preprint, 2020.
> >
> > > **Reviewer:** 4b. ...it may not be convincing that only the evaluation to the recovered attributes achieves higher performance. Besides, the sole evaluation metric is feature fidelity score, which is newly proposed in this paper. Therefore, the author should show more solid and fair experiment results.
> >
> > **Authors:** We believe that the problem introduced in the paper is an interesting and important one. As the setting itself is novel, there are no existing metrics or solutions. Since the setting itself is novel, and (as far as we know) there are no other contributions that try to recover the ground-truth attributes in an unsupervised manner, we have no direct comparison with other work that tries to optimize the overlap between the extracted features and a set of ground truth attributes. We do, however, perform evaluation of multiple baselines and show the advantage of our method in the newly introduced settings.
> >
> > We believe that the most solid experiments are those in which we add intersection regularization to a well established architecture and compare the results (last four rows of Tab. 1). Across all datasets we improve the feature fidelity while the degradation in accuracy is minimal.
> >
> > > **Reviewer:** 5. Lack of detailed implementation about the experiments including but not limited to the relevant data augmentation and training hyper-parameters such as batch size.
> >
> > **Authors:** In the experiments section, we provide the various loss coefficients,optimizer, stopping criterion, initialization and the evaluation process. We apologize for neglecting to mention some details. These are listed below and will be added to the paper.
> >
> > * Data Augmentations: we used the standard augmentations used in fine-grained classification tasks: (i) RandomHorizontalFlip with flipping probability p=0.5, (ii) RandomRotation with degrees divisible by 15 and (iii) RandomCrop with crop size 224.
> >
> > * We use a batch of size 64 across all of the experiments.
> >
> > * The learning rate is 0.001.

---

### Official Review · Reviewer_nTnf · 2021-11-23

**Confidence:** 4
**Overall Score:** 7

**Main Review:**

The paper proposes a clever and simple-to-implement way of learning interpretable representations for deep NNs, by jointly training them with decision trees. This question is relevant for the CleAR community since learning these types of representations is central to integrating causality with deep learning.

The work is technically sound, and clearly written. Related work is discussed in detail and the problem is well-motivated. The empirical experiments are convincing, and they are a nice demonstration that superior performance and interpretability do not necessarily go hand in hand. The ablation experiments also help strengthen the paper.

My only criticism is that when the implementation of the algorithm is presented, it is not clearly stated exactly which part of their pseudo-code and objective function corresponds to intersection regularization. Furthermore, it would be great if the authors can comment on whether the intersection regularization is actually necessary, if they were to use exactly the same architecture they proposed, and whether they have performed any experiments with that?

**Summary:**

This paper is concerned with learning interpretable deep representations by combining DNNs with decision trees.

---

> ### Author Response · Authors · 2021-12-03
> **Thank you for your review**
>
> Thank you for your supportive review, we appreciate the time and effort.
>
> > **Reviewer:** My only criticism is that when the implementation of the algorithm is presented, it is not clearly stated exactly which part of their pseudo-code and objective function corresponds to intersection regularization.
>
> **Authors:** The notion of intersection regularization is a generic idea that can be implemented in multiple ways with different variations. The concept of intersection regularization is applied between the functions $G$ and $T$ and is carried out in lines 7, 9, 11 and 12 in Algorithm 1. Specifically, in line 7 we update $G$ to minimize the error of $G \circ F$ with respect to the target function $y$, while being restricted to also be close to $T \circ F$ (e.g., $\min_{\theta} Q(\theta,\omega)$ in (4)). In ${S}’$ we aggregate samples for training the tree $T$ on top of $F$ (line 9). These are given by pairs $(F(x),G(F(x)))$. In lines 11-12 we essentially train $T$ to match the outputs of $G$ when their input is $F(x)$.
>
> > **Reviewer:** Furthermore, it would be great if the authors can comment on whether the intersection regularization is actually necessary, if they were to use exactly the same architecture they proposed, and whether they have performed any experiments with that?
>
> **Authors:** We experimented with the necessity of the proposed regularization. As we discussed in Section 6, paragraph ‘Ablation Study’, we conducted an experiment when training the same network architecture to minimize cross-entropy loss against the ground-truth labels (with and without the L1 constraint on the embedding layer). As depicted in Table 1, the fidelity scores of these variations (‘quantized net’ and ‘quantized net + L1’) are inferior to the fidelity performance of our full method.

---

### Decision · Program_Chairs · 2022-01-12

**Decision:**

Accept (Poster)

**Comment:**

This paper proposes a simple approach to learn interpretable representations. The approach is well positioned with respect to prior work and all reviewers found the paper interesting, well written, and technically sound.

It should be presented at the conference and will be of interest to the causality community.